# Carfilzomib’s Real-World Safety Outcomes in Korea: Target Trial Emulation Study Using Electronic Health Records

**DOI:** 10.3390/ijerph192013560

**Published:** 2022-10-19

**Authors:** Ha Young Jang, Hyun Kyung Lee, Chae Jeong Kim, Sung-Soo Yoon, In-Wha Kim, Jung Mi Oh

**Affiliations:** 1College of Pharmacy and Research Institute of Pharmaceutical Sciences, Seoul National University, Seoul 08826, Korea; 2Department of Internal Medicine, Seoul National University Hospital, 101, Daehak-ro, Jongro-gu, Seoul 03080, Korea

**Keywords:** real world safety, carfilzomib, relapsed/refractory multiple myeloma, target trial emulation

## Abstract

Carfilzomib is a promising anticancer drug for relapsed/refractory multiple myeloma (RRMM). However, real-world evidence has only investigated the cardiovascular safety of carfilzomib, and there is a high demand for thorough safety evaluations. We aimed to comprehensively evaluate the risk of adverse events associated with carfilzomib in Korean patients with RRMM. We followed up with 138 matched patients with RRMM (69 KRd (carfilzomib, lenalidomide, and dexamethasone) and 69 Rd (lenalidomide and dexamethasone) users). A total of 12 adverse events were evaluated. More than 75% of adverse events occurred during the early cycle (1–6 cycles), and the incidence rate showed a tendency to decrease in the later cycle (7–12 and 13–18 cycles). Severities of most adverse events were evaluated as grade 1-2. The KRd regimen were related with significantly increased risks of dyspnea (adjusted HR (aHR) 2.27, 95% confidence interval (CI) 1.24–4.16), muscle spasm (aHR 5.12, 95% CI 1.05–24.9) and thrombocytopenia (aHR 1.84, 95% CI 1.10–3.06). Although the severities were low, carfilzomib has many side effects in treating RRMM; hence, findings on the patterns of its adverse events could lead to both effective and safe use of KRd therapy in real-world settings.

## 1. Introduction

Multiple myeloma (MM) is a hematologic cancer caused by the abnormal differentiation and proliferation of plasma cells [1] and is the second most common hematologic cancer after leukemia [2]. Although the survival of multiple myeloma patients has at least doubled in recent years with the advent of a new paradigm, there are still unmet needs for the improvement of treatment outcomes [3,4]. Carfilzomib is a second-generation proteasome inhibitor [5] approved by the US Food and Drug Administration (FDA) in 2012 through an expedited review of patients with relapsed/refractory multiple myeloma (RRMM) [6]. The survival rate of RRMM dramatically increased after the introduction of carfilzomib, and a KRd therapy (combination regimen of carfilzomib, lenalidomide, and dexamethasone) was listed as category 1 in the multiple myeloma guidelines of the National Comprehensive Cancer Network (NCCN) [7]. In South Korea, carfilzomib was approved for the treatment of RRMM in March 2017, and for reimbursement in February 2018 [8].

Because of its expedited review process, KRd therapy lacks safety evaluation data, and there is a high demand for the post-marketing safety evaluation of KRd therapy. To date, several studies of safety evaluations have been conducted. However, there were clear limitations to these studies. First, most real-world data (RWD) studies have investigated only the cardiovascular safety of KRd therapy because the incidence of cardiovascular adverse events was frequent in the carfilzomib group in various randomized clinical trials (RCTs) [9,10]. Cardiovascular adverse events related to carfilzomib were analyzed in several meta-analyses [11,12]. Studies using RWD have also been conducted to determine the safety of carfilzomib in clinical settings. Observational studies were performed using US multicenter electronic medical records (EMR) [13], US insurance claims data (Medicare) [14], and the FDA Adverse Event Reporting System (FAERS) database [15]. These studies did not provide any other safety information regarding various organ systems. Second, most of the studies of carfilzomib were performed on white populations [5,9,10,16,17], and the safety of minority populations were not discussed. There are only a few real-world safety studies for non-white populations [8,18,19,20]; however, they focused on the efficacy or cardiovascular safety only. The side effects of chemotherapy can vary greatly owing to differences in clinical settings and ethnicity. Certainly, further studies on the safety of the KRd treatment in Asian populations are very crucial.

Target trial emulation is a research method that establishes a hypothetical clinical trial protocol to find answers to clinical questions that are difficult to test with RCTs and reproduces it through observational data [21]. In this way, the validity of the association analysis between medications and adverse events can be increased [22,23]. Target trial emulation methods have been employed in pharmacoepidemiological studies to ensure the robustness of the study results [24,25]. We aimed to evaluate the risk of adverse effects from KRd therapy compared to Rd (a combination regimen of lenalidomide and dexamethasone) therapy in Asian patients with RRMM by applying the target trial emulation method using electronic health records (EHRs).

## 2. Materials and Methods

### 2.1. Data Source

Clinical data related to demographic characteristics and adverse reactions were collected using EHRs and the clinical data warehouse (CDW) of Seoul National University Hospital (SNUH), which is a tertiary general hospital in South Korea. The CDW was developed to meet the needs for the utilization of electronic records in data analysis. All personal information was deidentified. This study was approved by the SNUH Institutional Review Board (No. 2107-201-1239).

### 2.2. Study Design

The protocol of the hypothetical target trial was based on the seven important components of the target trial derived from a study by Hernan et al. [21], and then emulated using the data from EHRs. The target trial design was similar to that of the ASPIRE trial (NCT01080391) [9], whose intervention and control regimens are commonly used in South Korea. The target trial protocol is shown in Appendix A.

The study design is illustrated in Figure 1. The follow-up period was set from the start date of the drug administration to the end date of follow-up or the censoring date, whichever occurred first. The drug administration start date, adverse event occurrence date, study end date, and censoring date are defined in Appendix A. The end date of follow-up was set to 19 months, considering that the maximum duration of KRd therapy was 18 cycles (1 month per 1 cycle). Considering the chemotherapy holiday, the occurrence of adverse reactions was continuously monitored without setting a separate censoring gap.

### 2.3. Study Population

The study subjects were new users of the KRd or Rd regimen for RRMM at SNUH. Patients with RRMM aged ≥18 years who received the KRd or Rd regimen were included. Considering the reimbursement date of carfilzomib and lenalidomide in South Korea, the enrollment periods for the KRd and Rd groups were set from February 2018 to December 2020 and from March 2014 to December 2020, respectively. Patients who participated in clinical trials, received carfilzomib or lenalidomide as another therapy, used the treatment drug as maintenance therapy, and/or were unable to be followed up with or assessed through EHRs were excluded.

### 2.4. Exposures

The test group received KRd in a 28-day cycle, and the dose was 20 or 27 mg/m^2^ of carfilzomib twice a week, 25 mg of lenalidomide on days 1–21, and 40 mg of dexamethasone per week. On days 1 and 2 of the first cycle, 20 mg/m^2^ of carfilzomib was administered intravenously for 10 min as the starting dose, and if the patient tolerated it well, the dose was increased to 27 mg/m^2^ on day 8. Carfilzomib was discontinued after 18 cycles and only lenalidomide and dexamethasone were maintained. The control group received Rd on a 28-day cycle, and the regimen was 25 mg of lenalidomide on days 1–21, and 40 mg of dexamethasone once a week.

### 2.5. Outcomes

Among the adverse reactions that occurred frequently (>25%) and differed by >5% between the KRd and Rd groups, cardiac-related adverse reactions (dyspnea, hypertension, acute renal failure, heart failure, and ischemic heart disease) in the ASPIRE trial were selected. The selected outcomes were hematological adverse reactions, such as thrombocytopenia, and nonhematologic adverse reactions, such as diarrhea, fever, cough, upper respiratory tract infection, hypokalemia, muscle spasm, dyspnea, hypertension, acute renal failure, heart failure, and ischemic heart disease. The operational definition for each adverse reaction was set with diagnosis codes, medical records, nursing records, prescription records, clinical examinations, and imaging test results based on previous research reports and the National Cancer Institute’s Common Terminology Criteria for Adverse Events (NCI CTCAE) version 5.024 (Appendix A) [26]. The occurrence of adverse events was divided into three parts by cycle: 1–6, 7–12, and 13–18 cycles.

### 2.6. Covariates

Basic patient information including age, sex, number of prior regimens, creatinine clearance, and prior therapies was collected at the drug start date. For disease severity, the diagnosis code was collected and the International Staging System (ISS) was investigated on the drug start date. To identify the patient’s underlying characteristics, the patient’s history, including 18 conditions, was collected prior to the drug start date, according to the information collection criteria of the ASPIRE trial; these conditions included 8 concurrent conditions (major surgery, active infection requiring treatments, human immunodeficiency virus infection, active hepatitis B or C infection, other malignancy, peripheral neuropathy, ongoing graft-vs-host disease, and pleural effusion or ascites) and 10 cardiac conditions (myocardial infarction, heart failure, angina, coronary artery disease, ventricular arrhythmia, sick sinus syndrome, acute ischemia, conduction system abnormalities, hypertension, and diabetes) (Appendix A) [9].

### 2.7. Statistical Analysis

Propensity score (PS) matching and baseline characteristic analysis were performed using SPSS^®^ software (version 26.0; IBM^®^ SPSS Inc., Chicago, IL, USA) and Microsoft Excel 2016. The KRd and Rd group were 1:1 matched to make the baseline characteristics between the two groups similar. The matching variables were age, sex, and the number of previous treatment regimens. For the baseline characteristics, continuous values were calculated as mean ± standard deviation or median (range), and the significance of the results was evaluated using Student’s t-test or the Mann–Whitney U test. For categorical variables, the chi-squared test or Fisher’s exact test was used. Statistical significance was set at *p* < 0.05.

The risk of adverse events was analyzed using SAS 9.4 (SAS Institute Inc., Cary, NC, USA). The frequency of adverse events, incidence (event/100 patient-cycle), and hazard ratios (HRs) were calculated. The Cox proportional hazards regression was used to estimate the HR for each outcome with a 95% confidence interval (CI). The HR was adjusted for age.

A sensitivity analysis was performed to establish the robustness of our statistical analysis. The sensitivity analysis was conducted with a contemporary comparator [27,28] by setting the enrollment period of the Rd group to be the same as the enrollment period of the KRd group (from February 2018 to December 2020).

## 3. Results

### 3.1. Demographics

Among all the patients diagnosed with multiple myeloma, a total of 447 patients who received KRd (*n* = 161) or Rd (*n* = 286) chemotherapy regimens were identified (Figure 2). After excluding patients who did not meet the predefined inclusion criteria, the eligible study cohort included 263 patients (72 KRd and 191 Rd users). After PS matching, 69 KRd users were matched to 69 Rd users. No significant difference was found between the two groups in terms of sex, number of previous treatments, ISS stage, and concurrent conditions, including cardiac conditions (Table 1). However, a marginal difference was observed in different ages (*p* = 0.05). The median age of the patients was 64.4 years and the proportion of men was 57.2% (*n* = 79).

### 3.2. Risk of Safety Outcomes in KRd Users

The median (range) of the total number of cycles of treatment differed by 6 (1–18) cycles in the KRd group and 12 (1–18) cycles in the Rd group. The incidence rates of all adverse events are shown in Table 2. At least one adverse event occurred in 88.4% and 81.2% of patients in the KRd group and Rd group, respectively. Moreover, 66.7% of the KRd group and 56.5% of the Rd group experienced at least one cardiac-related adverse event. Most adverse events (78.8%), including cardiac-related adverse events, occurred within one to six cycles of treatment. Severities of most adverse events were evaluated as grade 1–2. The proportion of patients who discontinued treatment due to adverse events was 18.8% (13 AEs) and 21.7% (15 AEs) in the KRd and Rd group, respectively. Mortality rates were 1.4% (one death) in the KRd group, and 5.8% (four deaths) in the Rd group (Appendix A).

In a Cox regression model, the KRd group showed a higher risk of all adverse events, except diarrhea (Table 3). Significantly higher risks of dyspnea were observed in the KRd group (adjusted HR (aHR): 2.27; 95% confidence interval (CI): 1.24–4.16; muscle spasm (aHR: 5.12, 95% CI: 1.05–24.94); and thrombocytopenia (aHR: 1.84, 95% CI: 1.10–3.06)), which are also shown graphically in the Kaplan–Meier curves (Figure 3).

When the results of the original RCT (ASPIRE trial) and RWE were compared (Appendix A), cardiovascular disease-related side effects were similar in the direction and significance of the results (each risk (CI) in the RCT and RWE = 1.19 [0.70–2.00]: 1.3 [0.53–3.16] (acute renal failure); 1.59 [0.83–3.02]: 1.45 [0.61–3.48] (cardiac failure); and 1.28 [0.68–2.42]: 1.00 [0.34–2.92] (ischemic heart disease)). The result of the hematologic side effect, thrombocytopenia, was also confirmed to be similar to that of the RCT (risk (CI) in the RCT and RWE = 1.4 [1.02–1.94]: 1.84 [1.1–3.06]). Dyspnea and muscle spasms were newly observed adverse events whose risks were significantly increased in our study (RWE), whereas the risks were not significantly increased in the RCT. Most of the patients who suffered dyspnea and muscle spasms developed cardiac side effects: 100% (49 of 49) and 90.9% (10 of 11) of patients with dyspnea and muscle spasms, respectively.

### 3.3. Sensitivity Analyses

The first sensitivity analysis was performed using the contemporary comparator, which was set with the same index period in the KRd and Rd groups (Appendix A). Similar to the results of the previous analysis, the risk of adverse events was higher in the KRd group. The risks of dyspnea and thrombocytopenia were significantly higher in the KRd group (aHR: 2.48, 95% CI: 1.21–5.09 (dyspnea) and aHR: 1.98, 95% CI: 1.09–3.59 (thrombocytopenia)), whereas the risk of muscle spasm was not significantly higher in the KRd group.

## 4. Discussion

This study investigated the risk of adverse events in KRd compared to Rd in patients with RRMM using real-world tertiary hospital EHR data. To the best of our knowledge, this is the first retrospective study to comprehensively review the various side effects of the KRd regimen in Koreans. Moreover, we carefully applied a target trial emulation method, which is a promising approach for evaluating carfilzomib’s safety in a hypothetical RCT. To date, studies on the safety of carfilzomib in Asians are limited. As such, the safety analysis for Koreans is meaningful because research results can vary depending on race. There are other research studies which analyzed carfilzomib in Koreans; however, they have focused on efficacy rather than safety. Therefore, the safety concerns of KRd and its tolerability in Korean patients remained uncertain. Our study included 138 elderly patients (median age: 64.4 years), and the results showed that KRd therapy significantly increased the risk of dyspnea, thrombocytopenia, and muscle spasm.

The KRd regimen was associated with frequent adverse events. Over 80% of patients had at least one adverse event, and 75% of the events occurred in 1–6 cycles. The treatment discontinuation rate due to adverse events was 18.8%, indicating that severities of most adverse events were low. Despite of its high incidence rate of adverse events, it seems that the KRd regimen is tolerable, and it could enable RRMM patients to undergo multiple lines of therapy while retaining the maximum response. KRd therapy did not significantly increase most of the adverse events except for dyspnea, thrombocytopenia, and muscle spasms.

As the median number of treatment cycles of KRd was six, it is likely that the treatments were often discontinued in the early cycle (up to six cycles) due to the adverse events of KRd. In several other studies, including RCTs [9,14,15,18,19], most adverse events tended to occur after 15–150 days or at the completion of 3–6 cycles, showing a trend similar to the results of this study. Therefore, close monitoring of the early cycles of KRd administration is required. The incidence of adverse events was decreased in the late cycles of seven or more, which means that the adverse events were not affected by the cumulative dose of carfilzomib. Therefore, the continued use of KRd would be possible once it is well tolerated.

Many other studies have reported the incidence of heart failure to be 2%–6%, and carfilzomib use showed a significantly increased risk of heart failure [9,11,13,14,17]. In a study using Medicare, the risk of adverse cardiac events and dyspnea was higher in the carfilzomib group than in the non-carfilzomib group [14]. In our study, the incidence of heart failure was similar to that reported in other studies. However, the risk of cardiac adverse events (e.g., heart failure, ischemic heart disease) due to the KRd regimen was not significantly increased compared to that induced by Rd. Contrastingly, the risk of dyspnea was significantly increased by the KRd regimen. The differences in the results might be due to differences in ethnic issues or clinical settings. First, Asians are reported to have a lower risk of cardiovascular disease than other races [29]. Second, physicians in Korea reduced the dose of carfilzomib or discontinued it before heart failure occurred. Note that dyspnea, along with edema, syncope, and chest pain, is classified as a cardiovascular-related adverse event and prognostic symptom of heart failure [11,12,14]. This pattern was also distinct in this study; all 49 patients who complained of dyspnea also developed adverse cardiac events. Furthermore, muscle spasm was also a newly observed adverse event in our study (RWE), unlike its nonsignificance in the ASPIRE study (RCT). Similar to dyspnea, 10 of the 11 patients who suffered from muscle spasms developed adverse cardiac events. Likewise, dyspnea and muscle spasm could be important prognostic symptoms as indicators of adverse cardiovascular effects, such as heart failure.

However, these prognostic symptoms may not explain all mechanisms of cardiovascular adverse events. Thus far, the pathogenesis appears to be multifactorial. Carfilzomib is a proteasome inhibitor that disrupts the function of peripheral nerves and skeletal muscles [30,31]. Particularly, this mechanism appears to cause myocyte damage [32]. Various studies have reported that carfilzomib may directly affect coronary resistance, vascular tone, and reactivity [33,34,35]. Other studies have shown that endothelial effects are potential mechanisms rather than echocardiographic findings [36]. Carfilzomib is also known to induce renal failure and microangiopathy, which is also mediated by endothelial dysfunction [36,37,38]. In order to prevent serious side effects, careful monitoring will be required considering the known carfilzomib’s endothelial effects, including dyspnea or muscle spasm found in our study.

Thrombocytopenia is a frequent hematologic adverse event in multiple myeloma that can be exacerbated by the type of treatment administered [39]. In our study, patients administered KRd had a significantly increased risk of developing thrombocytopenia compared to those administered Rd. More than 50% of thrombocytopenic reactions occurred in cycle 1. Furthermore, the incidence rates in each cycle decreased as the number of cycles increased (42%, 27.6%, and 14.3% in cycles 1–6, 7–12, and 13–18, respectively). The platelet level often returned to normal after the end of each cycle, indicating that carfilzomib has no cumulative effect on thrombocytopenia. The same trend has also been observed in other studies. In a phase 2 clinical study in which carfilzomib was administered alone, the patient reached the nadir before day 8, and thrombocytopenia subsided within a few weeks [40]. Therefore, it seems that the risk of thrombocytopenia can be lowered by providing a chemotherapy holiday between each cycle of carfilzomib. It can be stated that the optimized strategy is not meant for discontinuation or change of the therapy, but for managing the adverse events carefully, thereby maximizing the therapeutic effect of KRd therapy.

For the severity grade of adverse events evaluated by using CTCAE, most were grade 1-2. Patients in the KRd group (43.5%) developed dyspnea, and most were dyspnea on exertion (CTCAE grade 1). In the ASPIRE study, the incident rate of dyspnea was 19.4%, which was much lower than that in the RWE. The outcome definition in the RCT was not clearly presented, and our research team had to make some assumptions, which might have led to differences in the incidence rates. Ethnic factors may also have been involved because over 95% of patients were white, and less than 1% were Asian in the RCT [9]. Seven adverse events showed higher incidence rates in the RWE, including dyspnea, hypertension, acute renal failure, cardiac failure, ischemic heart disease, pyrexia, and thrombocytopenia, whereas five adverse events (diarrhea, cough, upper respiratory tract infection, hypokalemia, and muscle spasm) showed lower rates.

The strengths of this study include the use of cohort data with a long follow-up period (sufficient to cover a maximum of 18 cycles) for the Asian population. Of particular interest, using EHRs, adverse events based on the patient’s verbal complaints and clinical laboratory values were sufficiently detected. Based on several studies and the CTCAE guidelines, operational definitions for each adverse event were established, and adverse events were detected based on clear criteria. This approach allowed us to manually study various side effects. Another strength of this study was the use of the target trial emulation method, which made it possible to implement an environment similar to an RCT while minimizing bias.

However, this study had several limitations. First, it was not possible to completely exclude confounding factors and evaluate causality owing to the limitations of this retrospective study. The limitations were minimized by applying target trial emulation, PS matching, adjusting the HR with covariates, and using an active comparator. However, residual confounding factors may still have been present in our analyses. Second, this study was conducted with a single-center design, and therefore, lacked external validity. Lastly, due to the nature of a low incidence of MM (about 3 per 100,000) [41], our study includes a small number of patients. Additionally, in the process of the PS matching, many study subjects were excluded. The trend of the risk of adverse events in this study and the ASPIRE trial was similar, but not significant. Note that most of the other research on the real-world safety of carfilzomib analyzed about 100 study subjects, which is similar to the number of our study subjects (*n* = 197, Italy [17]; *n* = 156, US [5]; *n* = 55, Korea [20]; *n* = 40, Korea [8]). Our study used a similar or larger number of study subjects to present the integrated safety results for carfilzomib. Furthermore, we have found some prognostic symptoms for its severe adverse events. We hope that our study results will enable the safe and effective use of carfilzomib. Further research with a sufficiently large sample size (e.g., with a multicenter design) should be conducted.

## 5. Conclusions

The findings of our study suggest that most adverse events occurred in the early cycle, and the use of KRd therapy was associated with an increased risk of dyspnea and thrombocytopenia. Since there was no cumulative adverse effect of KRd and the severity of adverse events were mild, the treatment could be continued if the patients can tolerate it. However, caution should be exercised when prognostic symptoms that could lead to serious cardiac adverse events are observed. Since RRMM patients often have a fine line between treatment response and adverse events, we believe that our findings could be helpful in using KRd therapy effectively for RRMM patients.

## Figures and Tables

**Figure 1 ijerph-19-13560-f001:**
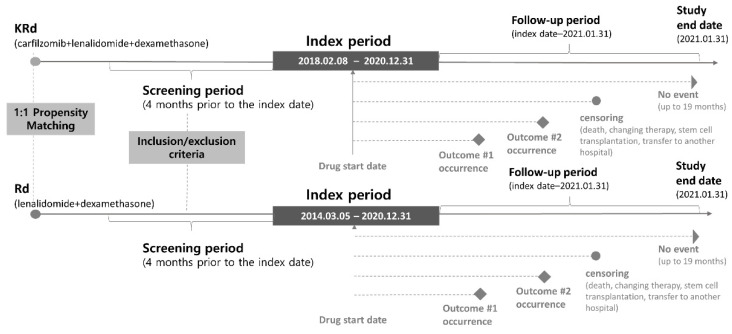
Flow diagram of safety analyses of KRd and Rd in relapsed/refractory multiple myeloma patients. KRd—combination regimen of carfilzomib, lenalidomide, and dexamethasone; Rd—combination regimen of lenalidomide and dexamethasone.

**Figure 2 ijerph-19-13560-f002:**
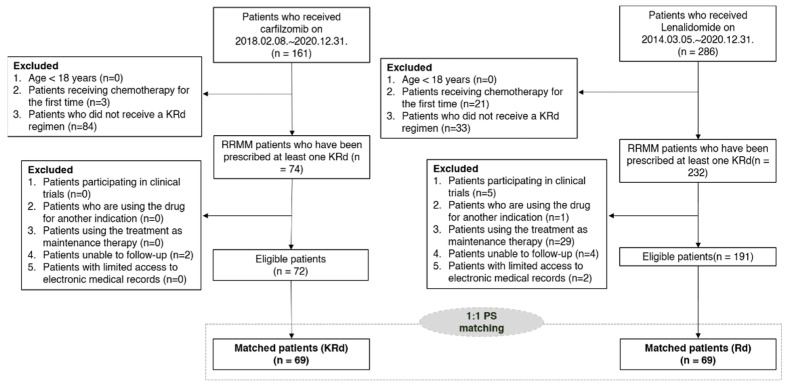
Flow chart showing experimental design. KRd—combination regimen of carfilzomib, lenalidomide and dexamethasone; PS—propensity score; Rd—combination regimen of lenalidomide and dexamethasone.

**Figure 3 ijerph-19-13560-f003:**
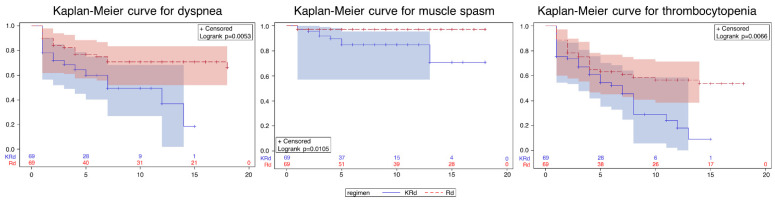
Kaplan–Meier curve for each adverse event. KRd—combination regimen of carfilzomib, lenalidomide, and dexamethasone; Rd—combination regimen of lenalidomide and dexamethasone.

**Table 1 ijerph-19-13560-t001:** Baseline characteristics.

	KRd (*n* = 69)	Rd (*n* = 69)	*p*-Value
Age—years ^‡^	64 (45–80)	67 (39–85)	0.04 *
Male sex ^†^	40 (58.0)	39 (56.5)	0.86
Number of prior regimens ^§^	2.0 ± 1.0	2.0 ± 1.2	0.76
Disease stage at initial diagnosis ^†^			
I	15 (21.7)	4 (5.8)	0.11
II	18 (26.1)	16 (23.2)	
III	13 (18.8)	13 (18.8)	
Unknown	23 (33.3)	36 (52.2)	
Creatinine clearance—mL/min ^§^	84.0 ± 39.4	60.1 ± 21.0	0.76
<50mL/min ^†^	14 (20.3)	19 (27.5)	0.32
≥50mL/min ^†^	55 (79.7)	50 (72.5)	0.32
Prior therapies ^†^			
Bortezomib	64 (92.8)	62 (90.0)	0.55
Lenalidomide	1 (1.4)	0	0.48
Any immunomodulatory agent	16 (23.2)	25 (36.2)	0.09
Concurrent conditions^†^			
Major surgery	0 (0)	2 (2.9)	0.24
Active infection requiring treatments	2 (2.9)	3 (4.3)	0.55
Human immunodeficiency virus infection	0 (0)	0 (0)	-
Active hepatitis B or C infection	7 (10.1)	4 (5.8)	0.35
Other malignancy	4 (5.8)	4 (5.8)	1.00
Peripheral neuropathy	30 (43.5)	34 (49.3)	0.50
Ongoing graft-vs-host disease	0 (0)	1 (1.4)	0.48
Pleural effusion or ascites	0 (0)	0 (0)	-
Cardiac conditions ^†^			
Myocardial infarction	0 (0)	1 (1.4)	0.48
Heart failure	2 (2.9)	4 (5.8)	0.40
Angina	3 (4.3)	1 (1.4)	0.31
Coronary artery disease	1 (1.4)	3 (4.3)	0.31
Ventricular arrhythmia	0 (0)	0 (0)	-
Sick sinus syndrome	1 (1.4)	1 (1.4)	1.00
Acute ischemia	0 (0)	0 (0)	-
Conduction system abnormalities	0 (0)	0 (0)	-
Hypertension	18 (26.1)	23 (33.3)	0.35
Diabetes	16 (23.2)	16 (23.2)	1.00

* Mann–Whitney U test; ^†^ no. (%); ^§^ mean ± SD; ^‡^ median (range); KRd—combination regimen of carfilzomib, lenalidomide and dexamethasone; Rd—combination regimen of lenalidomide and dexamethasone.

**Table 2 ijerph-19-13560-t002:** Incidence of adverse reactions by treatment cycle.

	KRd (*n* = 69)	Incidence Rate in Each Cycle (%)	Rd (*n* = 69)	Incidence Rate in Each Cycle (%)
Treatment cycles during specified period ^†^				
Cycles 1–6	69 (100.0)	-	69 (100.0)	-
Cycles 7–12	29 (42.0)		44 (63.8)	
Cycles 13–18	7 (10.1)		31 (44.9)	
Number of treatment cycles ^‡^	6 (1-18)		12 (1-18)	
Adverse event by treatment cycle ^†^				
Nonhematologic adverse events				
Dyspnea				
Cycles 1–6	25	(36.2)	16	(23.2)
Cycles 7–12	4	(13.8)	2	(4.5)
Cycles 13–18	1	(14.3)	1	(3.2)
Hypertension				
Cycles 1–6	17	(24.6)	19	(27.5)
Cycles 7–12	5	(17.2)	2	(4.5)
Cycles 13–18	1	(14.3)	5	(16.1)
Acute renal failure				
Cycles 1–6	9	(13.0)	9	(13.0)
Cycles 7–12	2	(6.9)	1	(2.3)
Cycles 13–18	0	0	0	0
Cardiac failure				
Cycles 1–6	9	(13.0)	9	(13.0)
Cycles 7–12	1	(3.4)	2	(4.5)
Cycles 13–18	1	(14.3)	0	0
Ischemic heart disease				
Cycles 1–6	5	(7.2)	7	(10.1)
Cycles 7–12	1	(3.4)	1	(2.3)
Cycles 13–18	0	0	1	(3.2)
Diarrhea				
Cycles 1–6	11	(15.9)	17	(24.6)
Cycles 7–12	5	(17.2)	2	(4.5)
Cycles 13–18	1	(14.3)	0	0
Cough				
Cycles 1–6	9	(13.0)	11	(15.9)
Cycles 7–12	2	(6.9)	2	(4.5)
Cycles 13–18	0	(0.0)	0	0
Pyrexia				
Cycles 1–6	23	(33.3)	15	(21.7)
Cycles 7–12	3	(10.3)	5	(11.4)
Cycles 13–18	2	(28.6)	1	(3.2)
Upper respiratory tract infection				
Cycles 1–6	9	(13.0)	5	(7.2)
Cycles 7–12	2	(6.9)	4	(9.1)
Cycles 13–18	0	0	1	(3.2)
Hypokalemia				
Cycles 1–6	6	(8.7)	3	(4.3)
Cycles 7–12	3	(10.3)	2	(4.5)
Cycles 13–18	0	0	0	0
Muscle spasm				
Cycles 1–6	8	(11.6)	2	(2.9)
Cycles 7–12	1	(3.4)	0	0
Cycles 13–18	0	0	0	0
Hematologic adverse events				
Thrombocytopenia				
Cycles 1–6	29	(42.0)	24	(34.8)
Cycles 7–12	8	(27.6)	3	(6.8)
Cycles 13–18	1	(14.3)	1	(3.2)

^†^ no. (%); ^‡^ median (range).

**Table 3 ijerph-19-13560-t003:** Hazard ratios for each adverse event.

	KRd (*n* = 69)		Rd (*n* = 69)		Adjusted HR ^¥^ (95% CI)
	Events	Events/100-Patient Cycle	Events	Events/100-Patient Cycle	
Nonhematologic adverse reactions					
Dyspnea	30	9.84	19	3.11	2.27 (1.24–4.16)
Hypertension	23	6.04	26	4.44	1.32 (0.73–2.41)
Acute renal failure	11	2.66	10	1.41	1.3 (0.53–3.16)
Cardiac failure	11	2.66	11	1.63	1.45 (0.61–3.48)
Ischemic heart disease	6	1.38	9	1.36	1 (0.34–2.92)
Diarrhea **^¶^**	17	4.51	19	3.13	1.02 (0.52–2.01)
Cough **^¶^**	11	2.95	13	1.99	1.06 (0.45–2.46)
Pyrexia **^¶^**	28	8.14	21	3.33	1.79 (0.99–3.26)
Upper respiratory tract infection **^¶^**	11	2.96	10	1.53	1.45 (0.59–3.57)
Hypokalemia **^¶^**	9	2.05	5	0.7	1.91 (0.62–5.88)
Muscle spasm **^¶^**	9	2.24	2	0.27	5.12 (1.05–24.94)
Hematologic adverse reactions					
Thrombocytopenia **^¶^**	38	12.71	28	5.12	1.84 (1.1–3.06)

^¶^ Hematologic and nonhematologic adverse events occurred more than 25% in the ASPIRE study with a difference of more than 5% between the two groups. ^¥^ HR was corrected for age. KRd—combination regimen of carfilzomib, lenalidomide and dexamethasone; Rd—combination regimen of lenalidomide and dexamethasone.

## Data Availability

The data presented in this study are available upon request from the corresponding author.

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
