# Peer review of "Carfilzomib’s Real-World Safety Outcomes in Korea: Target Trial Emulation Study Using Electronic Health Records"

_ijerph, 2022, doi:10.3390/ijerph192013560_

Round 1
Reviewer 1 Report
This real-world data study on KRd is excellently designed by using EMR and provides novel insights in the toxicities of this regimen with results being plausible. I recommend the acceptance without any changes in the manuscript.
Although the major advance of the study is target trial emulation according to the pivotal ASPIRE study, authors found that muscle cramps and thrombocytopenia are predictors of adverse events during KRd therapy. However, the authors used ICD10 to report AEs which is a major pitfall of the study due to not grading the severity of particular AE. I propose using Common Terminology Criteria for Adverse Events version V to grade the severity of AE and perform Cox regression analysis in terms of predictors.
Author Response
We were pleased to have an opportunity to revise our manuscript now entitled “Carfilzomib’s real-world safety outcomes in Korea: Target trial emulation study using electronic health records". In the revised manuscript, we have carefully considered reviewers’ comments and suggestions.
Please see the attachment.

Reviewer 2 Report
The authors studied relapse/refractory myeloma patient patients with KRD (carfilzomib, lenalidomide, and dexamethasone) verse RD (lenalidomide and dexamethasone) and concluded that dyspnea and muscle spasms might predict cardiac events. Though interesting, this real world data still have some issues to be solved.
1. The finding is not novel or not informative for clinical practice.
2. The statistic methods, the retrospective method still have bias for the results, with different enrollment period, not a randomized designed for patients to have KRD or RD, and also lack of information of treatment course and dose of each therapy.
2. The patient number is not much for conclusion, especially with many subgroup analysis.
3. Cardiac events have already know to patients with KRD therapy in trials and also real world.
Author Response

(The authors gave the same response as above.)

Reviewer 3 Report
1. The number of patients is small (69 vs 69) selected from a quite larger group of patients.
2. As dyspnoia is the main side effect, definition as also grading of dyspnoia should be provided, including specific exams used for evaluation. (In the aspire study 2,8% of patients develop grade >3 , or 19,4% all grades, 36% in the current study).
3. Median cycles of carfilzomib given were 6, meaning that 50% of patients discontinued the treatment before the cycle 6. Reasons for discontinuation should be provided (disease progression vs side effects).
4. As ASPIRE study is used as competitor, main differences should be clarified
a. Discontinuation in aspire was mainly to disease progression, and less than 1% due to side effects.
A significant higher percentage of patients in the current study developed dyspneia, as also ischemic heart attack even in control group (table 4, 36%).
5. The statement in introduction paragraph (line 3-4), that the response of myeloma patient is low (15-30%)) is not correct, especially nowdays by using the current protocols,) and should be changed.
Author Response

(The authors gave the same response as above.)

Reviewer 4 Report
Methodology in the abstract has not been written properly, and it is confusing
Author Response
Dear Editor and Reviewer,
We were pleased to have another opportunity to revise our manuscript now entitled “Carfilzomib’s real-world safety outcomes in Korea: Target trial emulation study using electronic health records". During the second round review, we have carefully considered reviewers’ comments and suggestions. The study highlight was clearly emphasized, and part of our manuscript was revised in detail. The manuscript has been much improved according to reviewers’ comment.
Please kindly check the revised manuscript and response letter.docx.

Reviewer 5 Report
In this study, Yang HY et al. analyzed the adverse events of KRd regimen in the real-world setting in Korea. I think this is somewhat interesting, but there are some problems in this manuscript as indicated below.
Major points
1. The authors demonstrated that the incidence of adverse events was not high in the late cycles of 7 or more. However, the median total number of cycles in the KRd group was 6 cycles. Among patients who could continue KRd more than 7 cycles, the proportion of patients who had adverse events of KRd in the late cycles of 7 or more does not seem low. The author should explain this more precisely.
2. In this study, 43.5% of patients in the KRd group developed dyspnea. I think that the incidence of dyspnea caused by KRd was too high. What do you think about this? In addition, the treatments were often discontinued in the early cycle due to the adverse events in the KRd group. Based on these data, I don’t think the KRd therapy is safe.
3. The authors demonstrated that muscle spasm could be prognostic symptom as an indicator for cardiovascular adverse events. However, cardiac events consist of various diseases, such as ischemic heart disease, myocardiopathy, arrhythmia, and so on. It seems difficult to explain these all conditions by only single symptom. What kind of mechanisms do you think of?
Minor points
1. This manuscript has too many revision histories. The authors should submit a clean manuscript with revision history deleted.
2. Page 1, line 3 in the Introduction section: I think the word ‘leukemia’ should be replaced to ‘lymphoma’.
I believe the authors must revise or at least discuss these points more precisely to publish in International Journal of Environmental Research and Public Health.
I hope that my comment is useful for the improvement of the article.
Author Response

(The authors gave the same response as above.)

Round 2
Reviewer 2 Report
Though the authors modified the manuscript as reviewers' suggestion accordingly, the study still have bias for the conclusion.
1. Thought the authors have matched method to adjust patient group, the retrospective study without randomized protocol and small number size compromised final results.
2. The finding should have more discussion about the possible mechanism and the prevention method for clinical benefit.
Author Response

(The authors gave the same response as above.)
